# Effect of Curcumin Nanoemulsions Stabilized with MAG and DAG-MCFAs in a Fructose-Induced Hepatic Steatosis Rat Model

**DOI:** 10.3390/pharmaceutics13040509

**Published:** 2021-04-08

**Authors:** Beatriz Agame-Lagunes, Peter Grube-Pagola, Rebeca García-Varela, Alfonso Alexander-Aguilera, Hugo S. García

**Affiliations:** 1UNIDA, Tecnológico Nacional de México, Instituto Tecnológico de Veracruz, Miguel Ángel de Quevedo 2779, Veracruz, Ver. 91897, Mexico; D18020001@veracruz.tecnm.mx; 2Instituto de Investigaciones Medico Biológicas, Universidad Veracruzana, Iturbide s/n, Veracruz, Ver. 91700, Mexico; pgrube@uv.mx; 3Tecnológico de Monterrey, Escuela de Ingeniería y Ciencias, Av. General Ramón Corona 2514, Nuevo México, Zapopan, Jalisco 45138, Mexico; rebecagv@tec.mx; 4Tecnológico de Monterrey, Escuela de Ingeniería y Ciencias, Ave. Eugenio Garza Sada 2501, Monterrey, N.L. 64849, Mexico; 5Facultad de Bioanálisis, Iturbide S/N, Col. Centro, Universidad Veracruzana, Veracruz, Ver. 91700, Mexico

**Keywords:** curcumin, hepatic steatosis, mono- and diacylglycerides, medium chain triglycerides, fructose, nanoemulsion

## Abstract

Current changes in diet, characterized by an increase in the intake of sweetened beverages, are heavily related to metabolic disorders such as non-alcoholic fatty liver. This condition can produce simple steatosis and, in worse cases, potentially result in steatohepatitis, fibrosis, and cirrhosis, comparable to the damage caused by the consumption of more or less 20–30 g of alcohol per day. The main objective of this research was to evaluate the effect of curcumin (*Curcuma longa*) nanoemulsions, using mono- and diacylglycerides medium chain fatty acids as stabilizers in an in vivo hepatic steatosis rat model. Pathology was induced by providing 30% fructose intake in the drinking water. Globule sizes under 200 nm that were stable for 4 weeks were obtained; curcumin encapsulated in the nanoemulsion was >70%. The results revealed an improvement regarding body and liver weight in the animals treated with curcumin nanoemulsions. A decrease in total cholesterol, LDL, AST/ALT, and HDL in serum was observed; however, no apparent improvement regarding serum glucose or triacylglycerides values was noted. Histological analysis showed a significant decrease in the extent of steatosis, inflammation, and brown adipose tissue in the treated animals.

## 1. Introduction

Hepatic steatosis is characterized by an accumulation of over 5% triacylglycerides in hepatocyte vesicles; this is a consequence of the increased uptake of fatty acids by the liver [1]. Triacylglycerides originate from dietary sources or by de novo lipogenesis. It has been shown that hepatic lipids in patients, with non-alcoholic fatty liver disease (NAFLD), originate in 14.9% of cases from inadequate diets, 26.1% from de novo synthesis, and 59% from adipose tissue [2]. NAFLD is considered the hepatic manifestation of metabolic syndrome, a serious health problem that affects between 30–40% of the male population and 15–20% of the female population worldwide with an increasing prevalence in people with obesity up to 60–80% [3,4,5].

Fructose intake has been considered a risk factor for the development of NAFLD, and when added to highly consumed beverages it can cause steatosis in only seven days [6]. Its consumption has increased compared to sucrose due to lower costs, transportability, and stability in a variety of industrially processed foods; consumption increased by 50% in 1990, compared to data from 1970 [7]. Fructose is a highly lipogenic monosaccharide, heavily used in the food industry; it is a sweetening agent mainly added in the form of high fructose corn syrup to a variety of foods and beverages, such as desserts and carbonated drinks [8]. Individuals that consume beverages sweetened with high fructose corn syrup (55% fructose) would ingest over 27 g of fructose per 500 mL [9]. By exposing murine models to ad libitum diets that include high concentrations of sucrose or fructose syrup, a predisposition for the development of cardiovascular disease and metabolic syndrome emerges; this is caused by fructose metabolism that produces the substrates for lipid synthesis [10]. Fructose consumption that exceeds 50 g/day has been recognized as an important risk factor for the development of metabolic syndrome [11].

It has been suggested that decreasing body weight, along with a healthy diet and/or exercise, improves insulin sensitivity and reduces intrahepatic triacylglycerides by up to 40% within two weeks [6]. Nevertheless, patients usually return to a sedentary lifestyle and bad eating. For this reason, alternative therapies have been pursued, i.e., the use of bioactive compounds such as curcumin (diferoylmethane), a polyphenolic phytochemical extracted from turmeric (*Curcuma longa*) rhizomes [12,13,14]. According to the FAO, curcumin consumption is safe at 1 mg/kg of body weight; however, clinical trials have been conducted to determine potential side effects by using doses of up to 12 g/day for 3 months, without any adverse effects reported [15]. Published data propose that low concentrations of curcumin provide an hepatoprotective effect [16], capable of preventing fructose-induced hyperlipidemia and hepatic steatosis. It can also decrease triacylglyceride levels, the expression of LXRa and SREBP-1c liver proteins, lipogenic enzymes including ATP-citrate lyase, Acetyl-CoA carboxylase (ACC), and FAS that protect the mitochondrial structure [17], reduce levels of TNF-α and IL-6 [18], block LDL oxidation, increase bile secretion and cholesterol excretion, suppress gene expression of cholesterol synthesis, and protect from fibrogenesis in murine models [19]. Curcumin may also aid in preventing obesity and the metabolic complications involved [20]; it inhibits the activation of NF-κB and decreases ICAM-1, Cox-2, and MCP-1 in murine models with steatohepatitis; it has been reported to decrease fatty acid synthesis and activate acyl Co-A oxidase in simple steatosis, preventing lipid accumulation [21]. However, curcumin exhibits a very low solubility in water, <8 µg/mL, at alkaline or neutral pH. Fortunately, it has been demonstrated that the presence of lipids increases curcumin solubility [15]; for this reason, lipid nano-carrier systems have been studied as vehicles to improve solubility and increase bioavailability.

Nanoemulsions (NE) are transparent/translucent heterogeneous systems composed of a continuous phase and a dispersed phase in the form of droplets [22,23], characterized by particle sizes between 20–200 nm; however, not all authors agree on this range, this is because the size is considered to vary depending on the source, without being tightly specified [24]. Despite the droplet size, the differences between emulsion, microemulsion, and NE often generate confusion; NE mainly differ from emulsions in terms of gravitational stability, i.e., the smaller the size the greater the stability; evidence also shows that the bioavailability of lipophilic compounds increases when globule sizes are <200 nm. Additionally, NE differ from microemulsions, because of their smaller size and thermodynamic stability; nonetheless, they are adversely affected by changes in temperature and composition; therefore, NE are widely used in the food and pharmaceutical industries, even though they are thermodynamically unstable, meaning that eventually the phases will separate because of free energy difference [24,25,26]. To improve the stability of NE, structured emulsifiers such as monoacylglycerides (MAG) and diacylglycerides (DAG) enriched with medium-chain fatty acids (MCFAs) have been proposed, which not only provide stability to the system, but also confer health benefits to people with several metabolic disorders [27]; these emulsifiers are considered non-ionic small surfactant molecules commonly employed in food emulsions and as cosmetic and pharmaceutical emulsifiers. These acylglycerides are commonly produced by chemical processes using materials such as fats, oils, and glycerol that provide options to select the length of the chains that form them, providing a high impact on their functionality [28]; an example is CAPMUL^®^, which can be used as a primary solubilizer or as an emulsifier. MAG and DAG are produced by interesterification of oils with glycerol, forming a mixture of MAG, DAG, TAG, glycerol, and free fatty acids [28], thus allowing for better contact of the substrates without the need of a solvent [29].

The main objective of this research was to study the hepatoprotective effect of curcumin in an NE system stabilized and enhanced with a non-commercial emulsifier of mono- and diacylglycerides, enriched with medium chain fatty acids on an induced hepatic steatosis by high fructose intake on a Wistar rat model.

## 2. Materials and Methods

### 2.1. Materials

Glycerol from Golden Bell (Mexico City, México), 3 Å molecular mesh, Kolliphor^®^ EL from Sigma-Aldrich (Mexico City, México), caproic acid (purity ≥ 99.5%), caprylic acid (purity ≥ 98%), and capric acid (purity ≥ 98%) were purchased from Sigma-Aldrich (Mexico City, México). For the quantification of acylglycerides, chlorotrimethylsilane (TMC), hexamethyldisilazane (HMDS), and pyridine were also obtained from Sigma-Aldrich (Mexico City, México). Novozym 435 enzyme (Lipase from *Candida antarctica* fraction B) from NOVO (Salem City, VA, USA). Solvents used were of HPLC grade from Tecsiquim (Mexico City, México). Curcumin with a purity ≥98% was bought from LKT Laboratories (St. Paul, MN, USA). Swanson medium chain triacylglycerides (MCTs) form Fargo (ND) and Milli-Q water (Milli-Q corp. Bedford, MA, USA) were used to prepare the NE. A Zetasizer Nano- ZS90 dynamic light scattering device (Malvern Instruments Inc., Worcestershire, UK) was used to characterize the nanoemulsions (NEs).

### 2.2. Preparation of MAG and DAG Emulsifier Enriched with Medium Chain Fatty Acids

Glycerolysis reactions were performed without the use of solvents according to the method reported by Esperón-Rojas et al. (2017) [29]. Briefly, the substrates employed to prepare the emulsifier were glycerol, caproic acid (C6:0), caprylic acid (C8:0), and capric acid (C10:0); all three medium chain fatty acids (MCFAs) were in a 1:1:1 *w*/*w*/*w* ratio. Additionally, 10% of 3 Å molecular sieves (w% with respect to total substrates) and 5% of the immobilized lipase Novozym 435 (w% with respect to total amount of substrates) were added. The protocol was carried out in 25 mL Erlenmeyer flasks, and the reaction mixture was placed in a Thermo MaxQ 4450 orbital shaker set at 300 rpm and 50 °C for 30 min. Afterwards, chloroform was added to dilute the mixture, then micro-filtered to recover the immobilized enzyme and the molecular sieves. Finally, it was rotary evaporated and flushed with nitrogen, then stored in a freezer at −18 °C for further gas chromatography analysis.

### 2.3. Quantification of Acylglycerides

A gas chromatograph was used to quantify the acylglycerols. A total of 1 mL of pyridine, 0.2 mL of hexamethyldisilazane (HMDS), and 0.1 mL of chlorotrimethylsilane (TCS) were added to 100 mg of the reaction mixture; it was then vortexed and incubated at 40 °C in a dry block incubator for 15 min; later, the mixture was dried under nitrogen flow, and finally 1 mL of hexane was added, shaken, and centrifuged at 2000 rpm for 10 min; 0.1 µL were injected into a HP 5890 gas chromatograph fitted with a flame ionization detector and a PE-5 capillary column (30 m × 0.32 mm × 1 µm). The operating conditions were as follows: temperature of injector and detector 300 °C and 315 °C, respectively; the initial column temperature of 100 °C was held for one minute and raised to 300 °C at a rate of 10 °C/min and maintained for 20 additional minutes.

### 2.4. Preparation of Nanoemulsions (NE)

Oil/Water (O/W) curcumin NE were formulated using the previously obtained emulsifier of MAG and DAG enriched with medium chain fatty acids (MCFAs), as previously reported by our research group (Esperón-Rojas et al., 2020) [27], with slight modifications. Briefly, a proportion 3:97 O/W phases were used and prepared separately. The oil phase consisted of 2.5 mg/g of nanoemulsion of curcumin and medium chain triglycerides at 3% in the NE. The aqueous phase consisted of 49% Milli-Q water, 50% glycerol, and 1% emulsifier; 1% Kolliphor^®^ EL, was employed to dissolve curcumin in the oil phase. Once both phases were ready, they were individually placed in a Barnstead/Labline Aquawave 9376 (Singen, Germany) ultrasound bath for 30 min, then subjected to an Ultra-Turrax T25 homogenizer for 3 min at 20,000 rpm at 1 min intervals to form the coarse emulsion; to obtain the final NE, this was followed by ultrasonication at 20% amplitude in a Branson Digital S-450D ultrasonicator for 3 min, with 1 min intervals using an ice-water bath to avoid overheating the sample and avoid over-processing.

### 2.5. Characterization of Curcumin NE

A Zetasizer Nano S90 dynamic light scattering equipment (Malvern Instruments Inc., Worcestershire, UK), which uses backscattering technology and utilizes optics with a scattering detector angle of 90° at 25 °C to determine the average droplet size and distribution by PDI, was used. A 1:200 dilution of NE: deionized water was placed in a Zetasizer electrophoretic cell to avoid scattering effects. All samples were processed in triplicate.

### 2.6. Determination of Concentration and Entrapping Efficiency of Curcumin

The curcumin concentration was determined by HPLC as reported by Ochoa-Flores et al. (2017) [30]. Briefly, a standard curve was prepared with known curcumin concentrations dissolved in ethanol; these were centrifuged at 4000 rpm and 5 °C for 10 min. The supernatant was filtered by removing the material that was not solubilized; 5 µL of the supernatant was placed in 13 mm × 100 mm glass tubes and mixed with 5 mL of ethanol; 10 µL was injected into a Waters HPLC system equipped with a 600-quaternary pump, a 717plus autosampler, and a 2487 UV-Visible Detector fitted with an Alltech Econosphere™ C18 column (5 µm × 250 mm × 4.6 mm). The mobile phase consisted of acetonitrile, 2.8% acetic acid, and methanol at the concentrations described in Table 1, with a flow rate of 1 mL/min, set at a 428 nm wavelength. The retention time for curcumin was 4.6 min. The concentration of curcumin in the NE was determined from the final curcumin concentration after preparation and centrifugation.

The following equation was used to determine the entrapment efficiency: %EE = (MAC/TAC) × 100. Where %EE is the entrapping efficiency of curcumin in the NE, MAC is the curcumin measurement used in the preparation of the NE (mg/g NE), and TAC is the total amount of curcumin used in the NE in mg/g.

NE were stored at 4 °C to analyze changes in mean particle size and PDI.

### 2.7. NE Stability

Stability was evaluated in a Nano-Zs90 instrument (Malvern Instruments Inc., Worcestershire, UK) at a 90° fixed angle at 25 °C for four weeks at 4 °C to determine the average particle size (the intensity-weighted mean diameter of the scattering), which was calculated from the data of signal intensity vs. droplet diameter and PDI; the samples were diluted in a 1:200 ratio in deionized water to prevent multiple scattering effects.

### 2.8. NAFLD Induction in a Wistar Rat Model

The protocol was carried out in accordance to the methodology proposed by Aguilera et al. (2017) [31] with modifications. A total of 16 male Wistar rats of 21 days of age (kindly donated by the Universidad Cristobal Colón) were kept at 12 h light/dark cycles and 25 °C. The handling and maintenance of the animals was performed according to the National Research Council Guide for the Care and Use of Laboratory Animals, Eighth Edition (2011). The experimental protocol was approved by the institutional research/bioethics committee under the code CI-ITVER/10/2018 (26 October 2018).

The animal adaptation period was set for one week prior to separation. The experimental groups (n = 4 in each group) were divided as follows: one healthy control group (HC) and three experimental groups: sick control (SC), curcumin NE with MAG and DAG MCFAs (cNE-MCFA), and NE with MAG and DAG MCFAs without curcumin (NE-MCFA). All groups received standard diets (Laboratory Rodent Diet 5001, LabDiet); however, the experimental groups had 30% fructose added to their available drinking water for 10 weeks. Treatments were administered orally by gastric cannula (20 mg of curcumin/kg bw), and the control groups (sick and healthy) had 50/50 water/glycerol for two weeks after the 10 weeks of induction. Animals were fasted 18 h prior to sacrifice.

### 2.9. Serum Parameters

Blood samples were carefully collected by cardiac puncture to avoid hemolysis and centrifuged at 15,000 rpm for 10 min. Serum and plasma were stored at −20 °C for further analysis. Glucose, cholesterol, triglycerides, aspartate aminotransferase (AST), and alanine aminotransferase (ALT) were measured using spectrophotometric techniques.

### 2.10. Histological Analysis

To determine liver damage caused by fructose consumption, liver tissue and fat samples were placed in 10% formaldehyde to be stained with hematoxylin-eosin dyes. Images were captured using an Olympus BX51 microscope (Tokyo, Japan) at 10× and 40× enlargements, equipped with a digital camera.

### 2.11. Statistical Analysis

The Minitab^®^ 18 Statistical Software from Minitab Inc. (State College, PA, USA) was employed. The analysis of variance and paired comparisons using Tukey’s test (*p* < 0.05 or *p* < 0.01) were established at a confidence level greater than 95% to be considered statistically significant, and the results were expressed as mean ± standard deviation.

## 3. Results

### 3.1. Obtaining MAG and DAG Emulsifier from MCFAs

For MAG and DAG-MCFAs synthesis, the methodology proposed by Esperón-Rojas et al. (2017) [29] was followed, using 5% of Novozym 435, responsible for incorporating C6:0, C8:0, and C10:0 MCFAs into the glycerol molecule at 50 °C for 30 min. Afterwards, chloroform was added and filtered in order to recover the emulsifier. The reaction mixture was concentrated by rotary evaporation to obtain only the emulsifier; headspace was saturated with nitrogen to avoid fatty acid oxidation. Figure 1a depicts the conformation of the fatty acids in the mixture before starting the glycerolysis reaction. According to results obtained by gas chromatography analysis, a mean of 81.85% of the MAG and DAG mixture was obtained after 30 min of reaction (Figure 1b).

### 3.2. Curcumin NE with MAG and DAG

Emulsions were prepared in a 97:3 water-oil phase ratio with 1% emulsifier and 1% Kolliphor^®^ EL to dissolve curcumin as reported by Esperón-Rojas et al. (2020) [27]. A modification to the proposed methodology was the sonication time; it was observed in preliminary runs, that subjecting the sample to 3 cycles instead of 2 was favorable in decreasing particle size. This variation produced a particle size of 184.4 nm ± 1.02 nm. The comparison of the particle size and PDI of this work with other investigations using MCFAs are presented in Table 2.

### 3.3. NE Stability

The stability of the nanoemulsions, stored at 4 °C for 4 weeks, was determined using the Zetasizer equipment to determine variation in size or the presence of destabilizing mechanisms since no apparent physical change was observed. Condensed results are depicted in Figure 2.

### 3.4. Entrapment Efficiency of Curcumin in NE

A calibration curve was prepared with the known concentrations of curcumin in ethanol; these were injected into a Waters HPLC system fitted with an Econosphere C18 column (5 µm, 250 mm × 4.6 mm). Curcumin was detected at 428 nm with a retention time of 3.0 min, according to the method reported by Ochoa-Flores et al. (2017) [30]. A curcumin entrapment of 76% in the formulated NE was calculated.

### 3.5. Experimental Animal Model

Animal subject data from all four groups, regarding body weight, water and fructose ingestion, feed intake, total energy consumption, and liver weight during the full 10 weeks of steatosis induction plus two additional weeks for the corresponding treatments are described in Table 3. There was no significant difference in body weight at the beginning of the protocol; however, at the end of the experimental period a significant difference was observed between the HC group, with a 54% increase, and the cNE-MCFA group, with a a 32% increase; meanwhile, the SC and NE-MCFA groups showed no difference between them despite having increased their body weight by 38% and 43%, respectively. The consumption of fructose contained in the drinking water was higher during the first week in the SC (78.83 ± 2.25 mL/day), NE-MCFA (76.00 ± 7.94 mL/day), and cNE-MCFA (80.00 ± 6.54 mL/day) groups, unlike the HC group (49.50 ± 6.54 mL/day). However, by the end of the protocol, the consumption was higher in the HC group (106.75 ± 2.25 mL/day). Concerning total food consumption, a significant difference was observed between the HC group which had the highest intake (19.13 ± 4.89 g/day/100 g bw) and the cNE-MCFA group with an apparent but not significant decrease in feed consumption. (10.97 ± 2.40 g/day/100 g bw) at the end of the experimental period. There was no statistical difference between the SC (13.60 ± 2.87 g/day/100 g bw) and NE-MCFA (14.09 ± 3.34 g/day) groups; the caloric consumption was significantly higher in the HC group when compared to the SC, NE-MCFA, and cNE-MCFA groups. The hepatosomatic index was higher in the SC group (4.79) compared to the NE-MCFA (2.73) and cNE-MCFA (2.45) groups. These results can be considered comparable to the HC group (2.58).

### 3.6. Serological Parameters

Serological parameters evaluated are depicted in Table 4; no significant differences in glucose values between groups were observed, despite an apparent increase in the cNE-MCFA group. Total cholesterol values decreased by >57% and >29% in the NE-MCFA and cNE-MCFA groups, respectively, compared to the SC group. However, triacylglycerides content in the cNE-MCFA group increased by 40% compared to the HC group and decreased in the NE-MCFA group by 31% compared to the SC group. HDL (26% and 13%), LDL (36% and 22%), and the LDL/HDL and cholesterol/HDL ratios improved in the NE-MCFA and cNE-MCFA groups. Regarding transaminase levels, the only group that showed a significant increase in both enzymes was the SC, with an AST/ALT ratio <1.

### 3.7. Histology

A histological analysis of both hepatic (Figure 3) and adipose (Figure 4) tissues was performed in order to assess the extent of steatosis in each experimental group. A semi-quantitative analysis of the lipid vacuoles was performed in 20 fields at 40×, as shown in Table 5. The criteria applied to determine if a field was considered positive were based on the presence of over 50% of lipid vacuoles. In hepatic tissue, the HC group did not exhibit alterations, while the SC group developed steatosis in 15% of analyzed tissue; the NE-MCFA group developed 10% steatosis, while the cNE-MCFA group developed 5% steatosis. In adipose tissue, mild chronic inflammation was observed in the SC as well as in the NE-MCFA and cNE-MCFA groups; aside from the inflammation, the cNE-MCFA group also contained brown adipose tissue; the HC group did not display any alterations.

## 4. Discussion

The purpose of the present research was to elucidate the effects of curcumin NE, stabilized with MAG and DAG synthesized from MCFAs, in an induced steatosis Wistar rat model by the dietary consumption of 30% fructose. The combination of the benefits from the thermogenicity of MCFAs and the hepatoprotective capacity of curcumin were pursued.

For several decades, an increase in the consumption of products with high fructose content, such as desserts and carbonated drinks, has been reported [9] to increase the risk of developing dyslipidemia, obesity, insulin resistance, and metabolic disorders [8]. It has been observed that adding fructose to drinks can cause steatosis in as little as 7 days [6]. In a study conducted by Lê et al. (2009) [32], it was described that adding 35% fructose to beverages and having an isocaloric diet increased liver lipid accumulation and decreased insulin sensitivity after 7 days in healthy subjects with a medical history of at least one parent with type 2 diabetes. Weight reduction through diet, exercise, or the use of therapeutic drugs, such as Metformin, has been suggested to counteract liver damage; however, the duration and effect of these therapies vary and are limited by patient endurance. There are multiple documented benefits of using natural products, with curcumin attracting great attention, and it is believed to have anti-inflammatory, anti-cancer, and hepatoprotective properties with low toxicity and side effects, compared to known drugs such as Metformin (dimethylbiguanide) used in the treatment of type 2 diabetes; however, the response obtained by metformin consumption does not cover all the clinical requirements of hepatic damage [17,20,33,34].

There have been several reports on the benefits of using lipids as bioactive compound carrier systems; the most frequently used include MAG and DAG and MCFAs. These provide effective compounds of interest for encapsulation and enhance their delivery to target cells or overall organs; this is achieved by forming more micelles in the gut compared to long chain fatty acids [15].

In this research, curcumin NE were prepared using MAG and DAG enriched with MCFAs, resulting in a final particle size of 174 nm. Our findings are comparable with those reported by Esperón-Rojas et al. (2020) [27], who obtained systems with particle sizes <200 nm (Table 2). In this research, protocol modifications yielded smaller particle sizes by performing three sonication cycles instead of two; globule size decreased by 10 nm. It has been reported that NE diameter can be reduced by increasing sonication time; similarly, the PDI remained in a range between 0.1 and 0.2, indicating stability as observed in Figure 2. It has been stated that producing a NE in the range of 20 to 200 nm provides benefits in terms of stability, a translucent appearance, bioavailability of lipophilic compounds, and a PDI of <0.5 indicates particle size distribution under a monomodal shape [35]. Entrapment efficiency (%EE) was determined to be 76%; nevertheless, this cannot be compared to data from Esperón-Rojas et al. (2020) [27] since quantification of encapsulated curcumin was not determined. A total of 100% curcumin entrapment in NE systems has been previously reported when using other emulsifiers such as phosphatidylcholine and different techniques [30]. In another study performed by Sadegh Malvajerd et al. (2019) [36], 94% of curcumin entrapment was achieved with the use of lipid carriers prepared with oleic acid; in this case, Tween was added to the oily phase, suggesting high affinity of curcumin with other emulsifiers.

In the Wistar rat model of induced steatosis by the addition dietary fructose, all animals showed constant weight gain, as opposed to information recorded by animal models where fructose was used to induce NAFL [37]. Test subjects did not gain a significant amount of weight when compared to controls. Nonetheless, after induction of steatosis for 10 weeks with 30% fructose in the provided drinking water, and two weeks with curcumin treatment in the corresponding experimental groups, a decrease in body weight was observed in the cNE-MCFA group with respect to the SC group. This information is consistent with results reported by Kelany et al. (2017) [38] who noted a decrease in body weight after 8 weeks in the experimental phase with 200 mg/kg-day of curcumin; also, in an investigation carried out by Um et al. (2013) [39], C57BL/6J mice were fed a high-fat diet for 11 weeks, obtaining a significant decrease in body weight in the group with curcumin at 0.15%, proving that the effect of body weight decrease does not only occur in the models induced with fructose, but also in those with a high-fat diet. However, no differences were found between the animals in the SC and NE-MCFA groups regarding weight and feed consumption as reported by Guimarães et al. (2019) [40] in 3-month-old C57bl/6 mice after 12 weeks, when fructose and medium chain triglycerides were administered as treatment. Additionally, in the SC, NE-MCFA, and cNE-MCFA groups, feed consumption decreased significantly compared to the HC group and was comparable to the data reported by Mamikutty et al. (2014) [41]. Water intake in the first weeks was higher in groups provided with 30% fructose; this can be attributed the high palatability of fructose. It is important to note that by week 12 there was a slight decrease. In a study conducted by Mamikutty et al. (2014) [41], 20% and 25% fructose were given in water for 8 weeks; it was concluded that the group with 25% fructose consumed less compared to the group with 20% fructose and the control group. Conversely, in terms of caloric intake (Kcal/day/100 g bw), the SC, NE-MCFA, and cNE-MCFA groups did not show significant differences amongst them. In the present work, an apparent decrease in total caloric intake (Kcal/day/100 g bw) was observed in the cNE-MCFA group, which can be attributed to a slight decrease in food consumption (g/day/100 g bw); additionally, this group reported the lowest body weight, confirming previous reports stating the beneficial effect of curcumin against obesity [42].

The use of animal models to assess the effects of fructose intake has been reported as a reliable predictor for the development of NAFL, attributed to rapid metabolism and passage through the liver to enter the glycolytic pathway [37]. Several adverse effects such as dyslipidemia, obesity, increased serum glucose, cholesterol, and triacylglycerides have been attributed to the excessive consumption of this monosaccharide. In this research it was determined that after 10 weeks of high fructose intake and two following weeks of curcumin treatment, there was no significant difference of serum glucose levels in any group, as reported by Francisqueti et al. (2016) [43]. In research conducted by Panahi et al. (2016) [44], patients diagnosed with NAFL were subjected to controlled curcumin intake (1 g/day) for 8 weeks; the results showed a decrease in cholesterol and triacylglyceride levels with no differences in fasting glucose levels. In the present study, triacylglyceride levels increased in the SC group compared to the HC and NE-MCFA groups; however, in the cNE-MCFA group a significant increase was observed despite a decrease in cholesterol levels. In most studies carried out to prove the beneficial effects of curcumin on serum parameters, these levels were diminished or unchanged. The results reported by Deters et al. (2003) [45] suggest that oral administration of curcumin at a 100 mg/kg bw produced no changes in cholesterol or triglyceride levels; nevertheless, a group treated with the same concentration of curcumin and cyclosporine did display an increase in these markers; this suggests the possible antagonistic effect that curcumin may have with some substances. In a report by Qiu et al. (2016) [46], two concentrations of curcumin were tested: 100 and 25 mg; at 100 mg, a decrease in body weight, cholesterol, GLUT 2, and COX 2 was observed; however, an increase in IL 6 and FAS occurred at the 25 mg concentration. The NE-MCFA group significantly decreased cholesterol and triacylglycerides levels in serum; these results are in accordance with data reported by Sung et al. (2018) [47] when comparing high- and low-fat diets with soy oil and medium chain triacylglycerides (MCT), attributing cholesterol and triacylglycerides reduction to increased activity of the lipolytic enzyme acyl-CoA oxidase, which contributes to enhanced β-oxidation. The rise in transaminase levels is the first diagnostic test for NAFL. Furthermore, an AST/ALT ratio < 1 usually suggests simple steatosis [48]. In this research the only group that reached this condition was the SC group, indicating that these animals developed liver damage caused by fructose intake.

It is well known that hyperlipidemia is linked to several diseases such as obesity, diabetes, chronic inflammation, and heart conditions. Curcumin has been reported to reduce the risk of atherosclerosis by regulating factors associated with inflammation and oxidative stress, including cytokines, protein kinases, and enzymes [49,50]. It has been shown that curcumin in a 0.02% *w*/*w* dose, as part of a daily diet, can protect against arteriosclerosis by moderating cholesterol levels and modifying lipoprotein concentrations responsible for regulating PPAR expression [49]. In this study, no significant changes were observed in HDL levels in the cNE-MCFA group. In a report published by Ghelani et al. (2019) [51], SD rats with adenine-supplemented diets and three different curcumin concentrations (50, 100, and 150 mg) for 24 days were evaluated; the data revealed that only the 100 and 150 mg/kg doses increased HDL levels, attributing the effect to the concentration of curcumin; however, the NE-MCFA group showed a slight increase in HDL levels. Zhang et al. (2016) [52] determined that the use of MCT increased HDL and decreased serum bile acid levels when compared to long chain triacylglycerides. Additionally, the same research group [53] had reported a decrease in TAG, LDL, and apolipoprotein levels after the use of MCFAs. It is important to note that in this research there was a significant decrease in LDL found in the NE-MCFA and cNE-MCFA experimental groups. This evidence suggests that both experimental groups were able to modify the weakening of lipoproteins metabolism. Several reports propose that the use of curcumin stimulates the excretion of cholesterol in the form of bile acids by stimulating CYP7A1 [51]; this produces an increase in lipase-sensitive hormone that facilitates lipolysis, it can also be increased by high levels of adipose triglyceride lipase in white adipose tissue; this may also accelerate the expulsion of cholesterol through feces excretion. Therefore, an increased cholesterol transport from peripheral tissues to the liver can be deduced [54]. An important cardiac risk prediction indicator is the increase in the cholesterol/HDL ratio [55]. The results obtained in this study show a significant decrease in both the NE-MCFA and cNE-MCFA groups when compared to the SC group; according to the American Heart Association (AHA), the ideal value should be 3.5.

A semi-quantitative analysis performed to describe changes in liver tissue and the presence of lipid vacuoles can be observed in Figure 4. In the cNE-MCFA group, brown adipose tissue was detected; this is attributed to curcumin intake that increases the levels of uncoupling protein UCP1, showing an intensification in adaptive thermogenesis. These browning effects have been proven to be mediated by the AMPK pathway and by the reduction in white adipose tissue inflammation [56,57,58,59]. Although it is known that MCTs are highly thermogenic and can generate brown adipose tissue, in this research said effect was not observed in the NE-MCFA group. Nevertheless, there was a decrease in fat deposits, probably caused by insufficient treatment administered that was unable to initiate the required signaling for conversion to fat browning, caused by the relatively short treatment period and low dosage.

Curcumin effects have been found to be dose-dependent in manner and are also highly influenced by the administration method caused by its low bioavailability. Multiple reports propose the hepatoprotective effect of curcumin; however, an effective dose has not yet been established in order to affect all parameters related to NAFL. Additionally, further research is required to determine the potential toxic effects that can be caused by the consumption of high curcumin concentrations. It has been suggested that excessive amounts can trigger oxidative stress, inflammation, and liver damage. In a clinical study published by Fernández-Aceñero et al. (2019) [60], after treatment intended for osteoarthritis, a female patient displayed abnormal biochemical parameters attributed to liver damage. When reviewing her medical history, the patient stated that she used curcuma as an adjuvant treatment; it was concluded that curcumin exerted an antagonistic effect with the drug used in the study (Etoricoxib) since both decreased COX2; it was determined that its use should be intermittent. Diverse formulations have been made trying to encapsulate curcumin in nanocarrier vehicles and use it in lower concentrations but with increased effectiveness. Jazayeri-Tehrani et al. (2019) [61] used a product called Sina curcumin^®^ by the Exir-NanoSina Company at a dose of 80 mg/day. The product consists of nanomicelles containing curcumin with sizes less than 100 nm in obese patients diagnosed with NAFL. Increased levels of HDL and a decrease in transaminase levels were observed, but no change in the weight of patients was recorded. Lee et al. (2019) [62] investigated the efficiency of turmeric nanoemulsions on liver damage in Balb/c mice after 9 weeks of administering turmeric extract (TE) or turmeric extract in nanoemulsion (TE-NE) at a concentration of 300 mg/day. The authors incorporated a lower curcumin content in the nanoemulsion; however, both systems had similar results in decreasing liver damage caused by palmitate ingestion.

## 5. Conclusions

The use of encapsulated curcumin in the NE carrier system, using MAG and DAG enriched with MCFAs, resulted in a good strategy to reduce some of the adverse effects of a high fructose diets in a Wistar rat model. It was determined that this system was capable of increasing the bioactivity of curcumin without the need for high concentrations, and possible side effects can be avoided. Further studies are required in order to better understand the metabolic pathways involved in the effects of curcumin with this emulsifier on steatosis, as well as the apparent synergism.

## Figures and Tables

**Figure 1 pharmaceutics-13-00509-f001:**
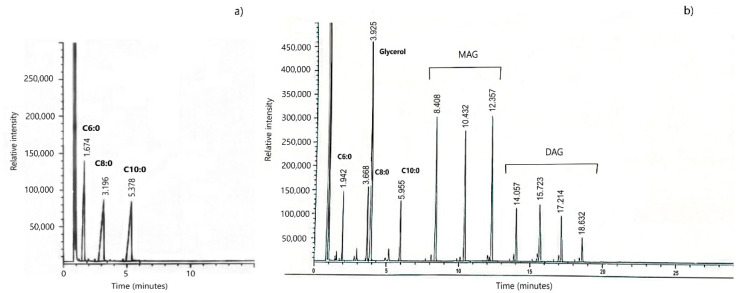
(**a**) Chromatogram of the medium-chain fatty acids (MCFAs) mixture (C6:0, C8:0, and C10:0) prior to glycerolysis reaction; (**b**) glycerolysis reaction products chromatogram with C6:0, C8:0, and C10:0 and glycerol (1:1 *w*/*w*) with 5% Novozym 435 and 10% of molecular mesh at 50 °C for 30 min.

**Figure 2 pharmaceutics-13-00509-f002:**
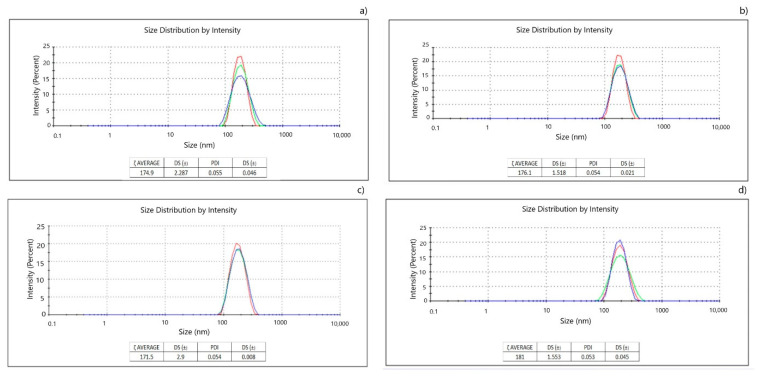
Particle size graphs provided by the Zetaizer Nano S90 equipment during 4 weeks of stability measurement of curcumin NE with monoacylglycerides (MAG) and diacylglycerides (DAG) enriched with MCFAs as emulsifier. (**a**) Week 1, (**b**) week 2, (**c**) week 3, (**d**) week 4. Separate measurements were performed in triplicate, indicated by the colored lines.

**Figure 3 pharmaceutics-13-00509-f003:**
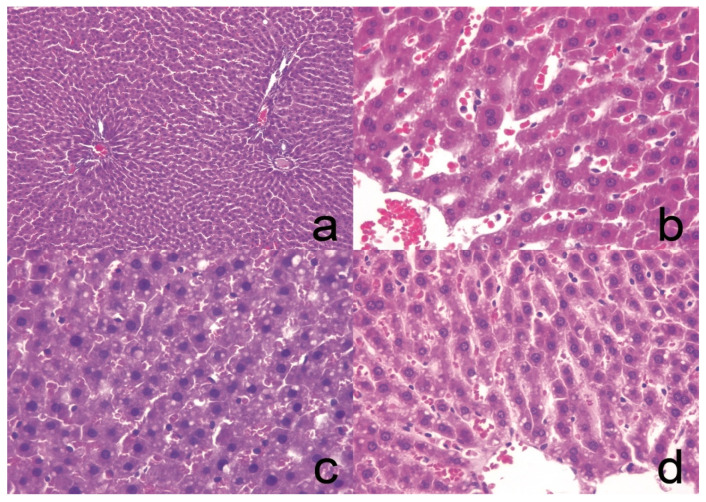
(**a**) Healthy control (HC) group liver, no histological changes observed (hematoxylin-eosin, 10×); (**b**) cNE-MCFA group liver tissue, small lipid vacuoles (steatosis) are identified in up to 5% of hepatocytes (hematoxylin-eosin, 40×); (**c**) SC group, higher numbers of lipid vacuoles were observed in up to 15% of hepatocytes (hematoxylin-eosin, 40×); and (**d**) NE-MCFA group, up to 10% of hepatocytes were identified to have lipid vacuoles (hematoxylin-eosin, 40×).

**Figure 4 pharmaceutics-13-00509-f004:**
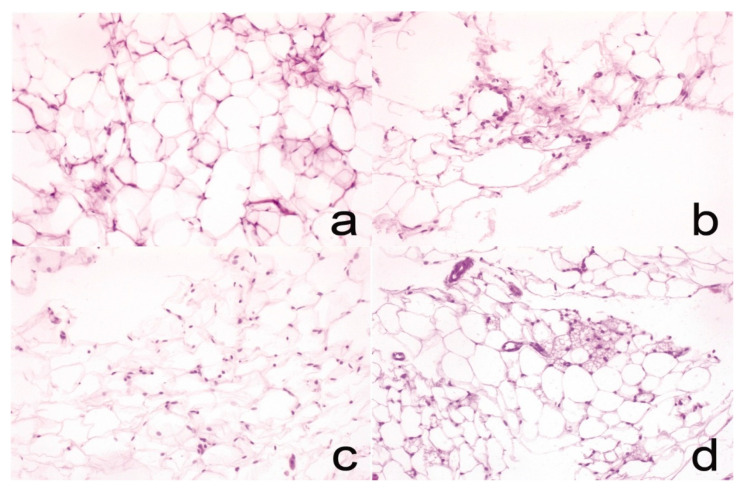
(**a**) HC group, adipose tissue without histological alterations (hematoxylin-eosin, 20×); (**b**) SC group, adipose tissue with lymphocyte isolates (hematoxylin-eosin, 20×); (**c**) NE-MCFA group, adipose tissue with low lymphocyte inflammatory infiltration (hematoxylin-eosin, 20×); (**d**) cNE-MCFA group, mixed adipose tissue, the presence of both brown and white adipose tissue was noted; the brown adipose presents microvacuoles while the white fat shows only one that covers the entire cytoplasm (hematoxylin-eosin, 20×).

**Table 1 pharmaceutics-13-00509-t001:** Mobile phase gradients for curcumin quantification.

Time	Acetonitrile (%)	2.8% Acetic Acid Solution (%)	Methanol (%)
0	35	55	10
4	45	25	30
6	40	30	30
8	35	55	10

**Table 2 pharmaceutics-13-00509-t002:** Particle size (nm) and PDI comparison of curcumin nanoemulsions (NE) using MCFAs as oil phase.

Author	Emulsifier	Mean Size (nm)	PDI
Esperón-Rojas et al., 2020	MAG and DAG MCFAs	184.4 ± 0.02	0.091 ± 0.002
Ochoa-Flores et al., 2017	PC-MCFAs	29.6 ± 2.3	0.24 ± 0.08
This research	MAG and DAG MCFAs	174.8 ± 1.11	0.195 ± 0.076

Data entries represent the means ± SD of experiments performed in triplicate.

**Table 3 pharmaceutics-13-00509-t003:** Body weight, water consumption, food intake, total caloric intake, and liver weight average after 10 weeks of 30% fructose consumption and two additional weeks of curcumin NE treatment.

Weight (g)	Healthy Control Group	Sick Control Group	NE-MCFA	cNE-MCFA
Starting weight	149.23 ± 9.40 ^a^	147.80 ± 7.44 ^a^	145.26 ± 1.68 ^a^	137.80 ± 2.67 ^a^
Final weight	331.16 ± 16.04 ^a^	240.43 ± 3.75 ^b^	256.2 ± 18.8 ^b^	204.1 ± 22.3 ^c^
Body weight gain	181.93 ± 13.95 ^a^	92.63 ± 7.79 ^b^	113.1 ± 17.4 ^b^	65.85 ± 12.76 ^c^
Liquid and food consumption				
Starting liquid intake (mL/day)	49.50 ± 6.54 ^a^	78.83 ± 2.25 ^b^	76.00 ± 7.94 ^b^	80.00 ± 6.54 ^b^
Final liquid intake (mL/day)	106.75 ± 2.25 ^a^	75.8 ± 18.5 ^b^	77.74 ± 4.98 ^b^	69.54 ± 8.96 ^b^
Liquid intake (mL/day/100 g bw)	31.23 ± 5.67 ^a^	43.50 ± 12.36 ^b^	38.06 ± 10.18 ^ab^	43.54 ± 11.38 ^b^
Kcal equivalent	0.00 ^a^	52.20 ± 14.83 ^b^	45.67 ± 12.22 ^b^	52.25 ± 13.66 ^b^
Food consumption (g/day/100 g bw)	19.13 ± 4.89 ^a^	13.60 ± 2.87 ^b^	14.029 ± 3.34 ^b^	10.97 ± 2.40 ^b^
Kcal equivalent	65.63 ± 16.77 ^a^	46.63 ± 9.85 ^b^	48.12 ± 11.47 ^b^	37.63 ± 8.24 ^b^
Total Kcal/day/100 g bw	65.63 ± 16.77 ^a^	98.83 ± 22.7 ^b^	93.79 ± 21.92 ^b^	89.88 ± 17.54 ^b^
Liver				
Liver weight (g)	8.55 ± 0.015 ^b^	11.52 ± 1.49 ^a^	6.99 ± 1.00 ^bc^	4.99 ± 0.84 ^c^
Hepatosomatic Index	2.58	4.79	2.73	2.45

Data means ± SD of the rats in each of the experimental groups. Healthy control group (n = 4), sick control group (n = 4), NE with MAG and DAG MCFAs without curcumin (NE-MCFA) group and curcumin NE with MAG and DAG MCFAs (cNE-MCFA) group (n = 4). Mean values within a row with different superscript letters differ significantly *p* < 0.05.

**Table 4 pharmaceutics-13-00509-t004:** Serological parameters in Wistar rats after 30% fructose intake and two additional weeks of curcumin NE treatment.

	Healthy Control Group	Sick Control Group	NE-MCFA	cNE-MCFA
Glucose (mg/dL)	93.49 ± 6.91 ^a^	85.87 ± 17.39 ^a^	78.12 ± 15.66 ^a^	110.421 ± 30.6 ^a^
Cholesterol (mg/dL)	62.84 ± 3.07 ^ab^	79.24 ± 6.56 ^a^	33.44 ± 5.01 ^c^	55.47 ± 9.96 ^b^
Triacylglycerols (mg/dL)	87.36 ± 8.43 ^b^	116.77 ± 10.55 ^a^	80.07 ± 0.38 ^b^	122.69 ± 13.67 ^a^
HDL (mg/dL)	20.3 ± 0.75 ^a^	17.3 ± 0.720 ^b^	21.8 ± 0.755 ^a^	19.6 ± 1.51 ^ab^
LDL (mg/dL)	48.8 ± 3.96 ^ab^	54.2 ± 5.85 ^a^	34.3 ± 2.64 ^c^	42.2 ± 4.57 ^bc^
HDL/LDL	0.417 ± 0.018 ^bc^	0.321 ± 0.025 ^c^	0.638 ± 0.027 ^a^	0.469 ± 0.079 ^b^
LDL/HDL	2.40 ± 0.106 ^b^	3.13 ± 0.246 ^a^	1.57 ± 0.066 ^c^	2.18 ± 0.399 ^bc^
Cholesterol/HDL	3.099 ± 0.266 ^b^	4.578 ± 0.201 ^a^	1.539 ± 0.283 ^c^	2.826 ± 0.394 ^b^
Aspartate aminotransferase (AST) (U/L)	91.06 ± 6.87 ^b^	121.92 ± 10.8 ^a^	87.72 ± 11.94 ^b^	75.86 ± 9.23 ^b^
Alanine aminotransferase (ALT) (U/L)	18.38 ± 2.05 ^bc^	137.69 ± 7.06 ^a^	23.77 ± 2 ^b^	12.2 ± 2.88 ^c^
AST/ALT ratio	4.95	0.88	3.7	6.21

Data means ± SD of the rats in each of the experimental groups. Healthy control group (n = 4), Sick control group (n = 4), NE-MCFA and cNE-MCFA (n = 4). Mean values within a row with different superscript letters differ significantly *p* < 0.05.

**Table 5 pharmaceutics-13-00509-t005:** Percentage of steatosis and inflammation found in histological sections of hepatic and adipose tissue stained with hematoxylin-eosin dye in rats after 30% fructose intake.

Liver Tissue	Healthy Control	Sick Control	NE-MCFA	cNE-MCFA
	0%	15%	10%	5%
Adipose tissue	No alterations	Mild chronic inflammation	Mild chronic inflammation	Mild chronic inflammation/Brown adipose tissue

Data obtained from quantitative analysis in 20 fields at 40×.

## Data Availability

Experimental data may be available upon request.

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
