# Peer review of "Effect of Curcumin Nanoemulsions Stabilized with MAG and DAG-MCFAs in a Fructose-Induced Hepatic Steatosis Rat Model"

_pharmaceutics, 2021, doi:10.3390/pharmaceutics13040509_

Round 1
Reviewer 1 Report
My comments are attached as .pdf file.

Author Response
Response to Reviewer 1 Comments
The presented manuscript describes fabrication of oil-in-water nanoemulsion composed of mono- and diacylglycerides medium chain fatty acids and Kolliphor® EL for encapsulation of curcumin in a fructose-induced hepatic steatosis rat model. In my opinion, the article presents a low level of scientific novelty, because it presents a formulation well known in the literature, As the authors themselves emphasize. Moreover, there is a huge chaos in the paper.
The detailed comments:
Point 1. The authors lack basic knowledge on that nanoemulsion formulations. Nanoemulsions are kinetically (unlike microemulsions which are thermodynamically) stable formulations. The Introduction lacks one large paragraph regarding the features of nanoemulsions, their advantages and the research hypothesis why they were used. Please insert such a paragraph and include the following works in it: Soft Matter 12 (2016) 2826; Pharmaceutics 12 (2020), 587; Advances in Colloid and Interface Science 287 (2021) 102318 where you can find the most important features of nanoemulsions; Perhaps, after studying and completing this basic knowledge, the authors will know better what system they got and how to test the stability of these formulations.
Response 1:
Thank you very much for your comments, but need to tell you that we are not new in this area We started publishing in nanoemulsions and bioactives since 2010 and we have contributed to the area so far with 23 papers in JCR journals, covering basic studies of Z-potential, emulsions characterization to application of this technology to carry bioactive lipophilic compounds into murine models. We know all the basic aspects the reviewer refers to and have made the type of studies requested, from electron microscopy to pharmacokinetic studies; we just find no reason to include these in this particular work, because the main thrust was decided to be the use of a carrier that is well-known in our group, to deliver low concentrations of the bioactive in a murine model of NAFL. Hence, although we highly respect his/her opinions about our experience or knowledge concerning nanotechnology, we disagree and would kindly refer him/her to the profile of the main corresponding author in google scholar, or any other scientific index, from 2010 to date. Alternatively, we can provide a list of our publications.
Upon the reviewer’s request, the information suggested has been included in the text and the introduction reads now as follows, from line 89:
“Nanoemulsions (NE) are transparent/translucent heterogeneous systems composed of a continuous phase and a dispersed phase in the form of droplets [22,23], characterized by particle sizes between 20-200 nm; however, not all authors agree on this range, this is because the size is considered to vary depending on the source, without being tightly specified [24]. Despite the droplet size, the differences between emulsion, microemulsion and NE often generate confusion; NE mainly differ from emulsions in terms of gravitational stability, i.e. the smaller the size the greater the stability; evidence also shows that bioavailability of lipophilic compounds increases when globule sizes are <200 nm. Additionally, NE differ from microemulsions, because of their smaller size and thermodynamic stability; nonetheless, they are adversely affected by changes in temperature and composition; therefore, NE are widely used in the food and pharmaceutical industries, even though they are thermodynamically unstable, meaning that eventually the phases will separate because of free energy difference [24-26]. To improve the stability of NE, structured emulsifiers such as monoacylglycerides (MAG) and diacylglycerides (DAG) enriched with medium-chain fatty acids (MCFAs) have been proposed, which not only provide stability to the system, but also confer health benefits to people with several metabolic disorders [27]; these emulsifiers are considered non-ionic small surfactant molecules commonly employed in food emulsions, cosmetic, pharmaceutical emulsifiers. These acylglycerides are commonly produced by chemical processes using materials such as fats, oils and glycerol; that provide options to select the length of the chains that form them, providing a high impact on their functionality [28], an example is CAPMUL® which can be used as a primary solubilizer or as an emulsifier. MAG and DAG are produced by interesterification of oils with glycerol, forming a mixture of MAG, DAG, TAG, glycerol and free fatty acids [28]; thus, allowing for better contact of the substrates without the need of a solvent [29].”
The main objective of this research was to study the hepatoprotective effect of curcumin in a NE system, stabilized and enhanced with a non-commercial emulsifier of mono- and diacylglycerides enriched with medium chain fatty acids, on an induced hepatic steatosis by high fructose intake, on a Wistar rat model.”
Point 2. So far, the aim of the work is chaotic and completely clear to the recipients of such a good journal as Pharmaceutics. Please provide one big Scheme to include the idea of the performed studies.
Response 2:
We thank you for your comment, we hope that the scheme presented here will clarify the research carried out.
Point 3. I don't really know what MAG and DAG (surfactant, oil phase) are. Please clarify this in the experimental section. I don't see any chemical basis where mono- and diacylglycerides medium chain fatty acids can act as an emulsifier.
Response 3:
Thank you for your suggestion, information has been added in the introduction section from line 101 in order to explain what MAG and DAG are:
“To improve the stability of NE, structured emulsifiers such as monoacylglycerides (MAG) and diacylglycerides (DAG) enriched with medium-chain fatty acids (MCFAs) have been proposed, which not only provide stability to the system, but also confer health benefits to people with several metabolic disorders [27]; are considered non-ionic small surfactant molecules commonly used in food emulsions, cosmetic, pharmaceutical emulsifiers. These acylglycerides are commonly produced by chemical processes using diverse materials such as fats, oils and glycerol; this provides them a difference in the length of the chains that form them, providing a high impact on their function at laboratory level [28], an example is CAPMUL® which can be used as a primary solubilizer or as an emulsifier. MAG and DAG are produced by interesterification of oils with glycerol, forming a mixture of MAG, DAG, TAG, glycerol and free fatty acids [28]; thus, allowing for better contact of the substrates without the need of a solvent [29].”
We are also aware of the use of monoacylglycerides and diacylglycerides in the food industry and the beneficial effect they provide since 1989 (DOI: 10. 1016/0005-2760(89)90350-0); likewise, our research team has been studying this type of non-commercial emulsifiers (DOI: 10.5650/jos.ess17010 DOI: 10.1080/10242422.2019.1646254 DOI: 10.5650/jos.ess18081 DOI: 10.1016/j.bcab.2020 .101638) from obtaining a good yield of emulsifier, their separation and the use of such emulsifiers in the improvement of the bioavailability of bioactive compounds. Modifications were also made to the experimental section as suggested starting on line 155.
“2.4. Preparation of nanoemulsions (NE)
O/W Curcumin NE were formulated using the previously obtained emulsifier of MAG and DAG enriched with medium chain fatty acids (MCFAs), as previously reported by our research group (Esperón-Rojas et al., 2020) [27], with slight modifications. Briefly: a proportion 3:97 O/W phases were used and prepared separately. The oil phase consisted of 2.5 mg/g of nanoemulsion of curcumin and medium chain triglycerides at 3% in the NE. The aqueous phase consisted of 49% Milli-Q water, 50% glycerol and 1% emulsifier; 1% Kolliphor® EL, was employed to dissolve curcumin in the oil phase. Once both phases were ready, they were individually placed in an Aquawave 9376 (Barnstead/Labline) ultrasound bath for 30 minutes, then subjected to an Ultra-Turrax T25 homogenizer for 3 minutes at 20,000 rpm at 1 min intervals, to form the coarse emulsion; to obtain the final NE, this was followed by ultrasonication at 20% amplitude in a Branson Digital S-450D ultrasonicator for 3 minutes, with 1-minute intervals using an ice-water bath to avoid overheating the sample and avoid over-processing.”
Point 4. Where are zeta potential results? What is surface load? The condition of DLS and zeta potential measurements should be described in detail, i.e, angle, detection, laser, number of scans, buffer? I recommend to provide a separate paragraph, e.g. “Size and zeta potential measurements”. Why the Authors provide a dilution 1:200? It makes absolutely no sense for nanoemulsion measurements. In this case, the sample has a completely different composition and a different formulation is measured.
Response 4:
Dear reviewer, since we started from a previous formulation as can be seen in the following article DOI: 10.1080/10242422.2019.1646254, the zeta potential data were omitted, however, it was considered that by maintaining a mean particle size, a PDI and stability similar to the one obtained in that research, the required parameters were accomplished.
As mentioned in the methodology section in 2.5. Characterization of curcumin NE, in line 169; “A Zetasizer Nano S90 dynamic light scattering equipment (Malvern Instruments Inc., Worcestershire, UK), which uses backscattering technology, utilizes optics with a scattering detector angle of 90° at 25 °C to determine the average droplet size and distribution by PDI, was used. A 1:200 dilution NE: deionized water was placed in a Zetasizer electrophoretic cell, to avoid scattering effects. All samples were processed in triplicate.”
The dilution of the sample was performed according to the guidelines established in the manual "Zetasizer Nano User Manual" in the section called "Sample preparation" where they recommend to do it depending on the type of sample to avoid multiple scattering it is worth mentioning that our research team has used dilutions as can be seen in the following articles DOI: 10. 2174/156720181366616160919141428 DOI: 10.1039/c7fo0093333j DOI: 10.1080/10242422.2019.1646254 DOI:10.2174 / 1389200221666200429111928 under the recommendations of the manual.
Point 5. The all experimental section is described in a very poor way. Please improve this section to better show the all instruments, chemical compounds and the protocols used.
Response 5:
We appreciate your comment; we have modified this section to clarify the document. It can be noted from line 118; however, some of the devices are described in the corresponding sections of the paper.
Point 6. These are the commonly used oil phases. Only the Kolliphor® EL (non-ionic surfactant) can perform the function of surfactant with emulsifying properties in this formulation.
Response 6:
As mentioned above, the use of MAG and DAG mixtures is considered as some of the most important emulsifiers in the food industry because they account for about 70% of food emulsifier production; commercial emulsifiers contain 45-55% MAG and 38-45% DAG. MAG can form stable dispersions because they are polymorphic and emulsifying activity is further increased by incorporating propylene glycol esters, sorbitan esters or lactic acid esters (Moonen et al., 2015; McClements, 2015)
Point 7. Figure 1 is completely unclear. Please provide better resolution of the chromatogram.
Response 7:
We appreciate your suggestion; the chromatogram is now displayed with a better resolution on line 247.
Point 8. Figure 2 is unacceptable. Please provide the original graphs from DLS measurements to show the nanodroplets changing in the time.
Response 8:
We acknowledge your suggestion; the original graphs as provided by the Zetasizer Nano S90 device can be found in the text on line 267, as well as its description on line 268.
Figure 2. Globule size graphs provided by the Zetasizer Nano S90 equipment during 4 weeks of stability of curcumin NE with MAG and DAG enriched with MCFAs as emulsifier. a) week 1, b) week 2, c) week 3, d) week 4.
Point 9. Table 1. I recommend to standardize the form of nanodroplets size presentation, i.e. with standard deviation and the same significant numbers. It looks chaotic and very unprofessional.
Response 9:
Dear reviewer, Table 1 only describes the mobile phase used for the quantification of curcumin in the HPLC equipment since gradients of each of the solvents are required.
Point 10. The Authors should describe in detail the determination method of encapsulation/solubilization efficiency of the curcumin in the obtained formulations. It could be interesting to learn, how the authors determined the encapsulation efficiency, in view of somewhat complex UV spectra - curcumin loaded nanoemulsions vs. mangostin solution vs. empty nanoemulsions?
Response 10:
We appreciate your comment; we believe that by modifying the text of the methodology, we made this section more understandable, it can be seen from line 118.
Reviewer 2 Report
The manuscript (pharmaceutics-1151465) entitled “Effect of curcumin nano-emulsions stabilized with MAG & DAG-MCFAs in a fructose-induced hepatic steatosis rat model” provided sound discussion with a reasonable set of experiment. Indeed, the manuscript still needed some revision for further improvement considering the following given suggestion.
- It is suggested to include the finding of other investigator exploiting curcumin/nanocurcumin to improve the condition in obesity and/or hepatic steatosis.
- It is suggested to incorporate the purity grade of curcumin in reagent section (2.1.) and modify the title of this section as “materials”.
- Esperón-Rojas et al. (2020) has also prepared and characterized the curcumin nanoemulsion and similar method was adopted by the author as well. I have query, the same formulation system also used by the author or it is totally different formulation developed and characterized by the author. If it is a different formulation then advised highlighting the superiority of your formulation compared to the formulation developed by Esperón-Rojas et al. (2020) in respect of droplet size, PdI, solubility of curcumin in 1 ml formulation and stability. The author has highlighted the comparative results in table 2 but the finding is not significant compared to the previous investigation. A similar line of the research reported by the previous investigator. In my opinion, the manuscript majorly revised by highlighting and significance of the current investigation in light of the comparative discussion of the following published literature.
- Lee EJ, Hwang JS, Kang ES, Lee SB, Hur J, Lee WJ, Choi MJ, Kim JT, Seo HG. Nanoemulsions improve the efficacy of turmeric in palmitate-and high fat diet-induced cellular and animal models. Biomedicine & Pharmacotherapy. 2019 Feb 1;110:181-9.
- Jazayeri-Tehrani SA, Rezayat SM, Mansouri S, Qorbani M, Alavian SM, Daneshi-Maskooni M, Hosseinzadeh-Attar MJ. Nano-curcumin improves glucose indices, lipids, inflammation, and Nesfatin in overweight and obese patients with non-alcoholic fatty liver disease (NAFLD): a double-blind randomized placebo-controlled clinical trial. Nutrition & metabolism. 2019 Dec;16(1):1-3.
- Um MY, Hwang KH, Ahn J, Ha TY. Curcumin attenuates diet‐induced hepatic steatosis by activating AMP‐activated protein kinase. Basic & clinical pharmacology & toxicology. 2013 Sep;113(3):152-7.
- Overall, formulation and characterization aspects are poorly described and need extensive revision to highlight the merits of the developed formulation system compared to previously developed curcumin nanoemulsion system.
- It is suggested to include the result in a form of image of Droplet size distribution, PdI, Zeta potential, and droplet morphology of nanoemulsion by Transmission electron microscopy (TEM)/ cryo-TEM.
- It is advised to include the drug release profile of developed formulation system through dialysis bag technique and/or everted-gut sac study to determine permeability profile of developed curcumin nanoemulsion system.
- The author has mentioned in his conclusion that the developed system is capable of increasing the bioavailability of curcumin, therefore, it is suggested to include the pharmacokinetic profile of the developed curcumin nanoemulsion system in a suitable animal model to confirm the robustness/superiority of the developed formulation system compared to the previous report.
Author Response
Response to Reviewer 2 Comments
The manuscript (pharmaceutics-1151465) entitled “Effect of curcumin nano-emulsions stabilized with MAG & DAGMCFAs in a fructose-induced hepatic steatosis rat model” provided sound discussion with a reasonable set of experiment. Indeed, the manuscript still needed some revision for further improvement considering the following given suggestion.
Point 1. It is suggested to include the finding of other investigator exploiting curcumin/nanocurcumin to improve the condition in obesity and/or hepatic steatosis.
Response 1:
Dear reviewer, we believe that with the modifications made in the text, because of the suggestions provided, we made a better comparison of our results obtained by contrasting them with other research on the hepatoprotective properties of curcumin.
Point 2. It is suggested to incorporate the purity grade of curcumin in reagent section (2.1.) and modify the title of this section as “materials”.
Response 2:
Thank you for your suggestion; the title of this section is already modified on line 119 of the manuscript. The percentage purity of the curcumin used has also been added on line 126 of the manuscript.
“2.1. Materials
Glycerol (Golden Bell, Mexico City); 3 Å molecular mesh, Kolliphor® EL, caproic acid (purity ≥99.5%), caprylic acid (purity ≥98%) and capric acid (purity ≥98%) were purchased from Sigma-Aldrich (Mexico City). For quantification of acylglycerides, chlorotrimethylsilane (TMC), hexamethyldisilazane (HMDS) and pyridine were also obtained from Sigma-Aldrich. Novozym 435 enzyme (Lipase from Candida antarctica fraction B) from NOVO (Salem, VA). Solvents used HPLC grade from Tecsiquim (Mexico City). Curcumin with a purity ≥98% was bought from LKT Laboratories (St. Paul, MN). Swanson Medium Chain Triacylglycerides (MCTs) form Fargo (ND) and Milli-Q water (Milli-Q corp. Bedford, MA) were used to prepare the NE. A Zetasizer Nano- ZS90 dynamic light scattering device (Malvern Instruments Inc., Worcestershire, UK) to characterize the NEs.”
Point 3. Esperón-Rojas et al. (2020) has also prepared and characterized the curcumin nanoemulsion and similar method was adopted by the author as well. I have query, the same formulation system also used by the author or it is totally different formulation developed and
characterized by the author. If it is a different formulation then advised highlighting the superiority of your formulation compared to the formulation developed by Esperón-Rojas et al. (2020) in respect of droplet size, PdI, solubility of curcumin in 1 ml formulation and stability. The author has highlighted the comparative results in table 2 but the finding is not significant compared to the previous investigation. A similar line of the research reported by the previous investigator. In my opinion, the manuscript majorly revised by highlighting and significance of the current investigation in light of the comparative discussion of the following published literature.
Lee EJ, Hwang JS, Kang ES, Lee SB, Hur J, Lee WJ, Choi MJ, Kim JT, Seo HG. Nanoemulsions improve the efficacy of turmeric in palmitate-and high fat diet-induced cellular and animal models. Biomedicine & Pharmacotherapy. 2019 Feb 1;110:181-9.
Jazayeri-Tehrani SA, Rezayat SM, Mansouri S, Qorbani M, Alavian SM, Daneshi-Maskooni M, Hosseinzadeh-Attar MJ. Nano-curcumin improves glucose indices, lipids, inflammation, and Nesfatin in overweight and obese patients with non-alcoholic fatty liver disease (NAFLD): a double-blind randomized placebo-controlled clinical trial. Nutrition & metabolism. 2019 Dec;16(1):1-3.
Um MY, Hwang KH, Ahn J, Ha TY. Curcumin attenuates diet induced hepatic steatosis by activating AMP‐activated protein kinase. Basic & clinical pharmacology & toxicology. 2013 Sep;113(3):152-7.
Response 3:
Dear reviewer, we are sorry for the confusion that caused that section of the manuscript. We have decided to modify it to clarify future misunderstandings, slight but concise corrections can be found in section “2.4. Preparation of nanoemulsions (NE)” on line 155 of the manuscript
“2.4. Preparation of nanoemulsions (NE)
O/W Curcumin NE were formulated using the previously obtained emulsifier of MAG and DAG enriched with medium chain fatty acids (MCFAs), as previously re-ported by our research group (Esperón-Rojas et al., 2020) [27], with slight modifications. Briefly: a proportion 3:97 O/W phases were used and prepared separately. The oil phase consisted of 2.5 mg/g of nanoemulsion of curcumin and medium chain triglycerides at 3% in the NE. The aqueous phase consisted of 49% Milli-Q water, 50% glycerol and 1% emulsifier; 1% Kolliphor® EL, was employed to dissolve curcumin in the oil phase. Once both phases were ready, they were individually placed in an Aquawave 9376 (Barnstead/Labline) ultrasound bath for 30 minutes, then subjected to an Ultra-Turrax T25 homogenizer for 3 minutes at 20,000 rpm at 1 min intervals, to form the coarse emulsion; to obtain the final NE, this was followed by ultrasonication at 20% amplitude in a Branson Digital S-450D ultrasonicator for 3 minutes, with 1-minute intervals using an ice-water bath to avoid overheating the sample and avoid over-processing.”
We also appreciate the suggested bibliography, which can be found now in section “4. Discussion” on the line 397-400 and 500-511; this research study was based on the formulation previously carried out by our colleagues (Esperón-Rojas et al., 2020) but we reproduced those conditions to confirm the previously reported data; however, we highlighted the hepatoprotective activity of curcumin in this novel system. We also emphasize the benefit provided by the use of mono- and diacylglycerides along with medium chain fatty acids, for these reasons we mention only some relevant aspects of the nanoemulsions and noted the biological activity conferred by this system.
397 line:
“also, in an investigation carried out by Um et al. (2013) [40], C57BL/6J mice were fed a high-fat diet for 11 weeks, obtaining a significant decrease in body weight in the group with curcumin at 0.15%, proving that the effect of body weight decrease does not only occur in the models induced with fructose, but also in those with a high-fat diet.”
500 line:
“Diverse formulations have been made trying to encapsulate curcumin in nanocarrier vehicles and use it in lower concentrations but with increased effectiveness; Jazayeri-Tehrani et al. (2019) [62] used a product called Sina curcumin® by the Exir-NanoSina Company at a dose of 80 mg/day. The product consists of nanomicelles containing curcumin with sizes less than 100 nm, in obese patients diagnosed with NAFL. Increased levels of HDL and a decrease in transaminase levels were observed but no change in the weight of patients was recorded. Lee et al. (2019) [63] investigated the efficiency of turmeric nanoemulsions on liver damage in Balb/c mice after 9 weeks of administering turmeric extract (TE) or turmeric extract in nanoemulsion (TE-NE) at a concentration of 300 mg/day. The authors incorporated a lower curcumin content in the nanoemulsion; however, both systems had similar results in decreasing liver damage caused by palmitate ingestion.”
Point 4. Overall, formulation and characterization aspects are poorly described and need extensive revision to highlight the merits of the developed formulation system compared to previously developed curcumin nanoemulsion system.
Response 4:
We appreciate your comment; we have modified the methodology to clarify the document, which can be found from line 118 onwards.
Point 5. It is suggested to include the result in a form of image of Droplet size distribution, PdI, Zeta potential, and droplet morphology of nanoemulsion by Transmission electron
microscopy (TEM)/ cryo-TEM.
Response 5:
Thanks for your suggestion. In this paper we have only included the average particle size and the PDI in order to compare it with the results previously described in the literature by our laboratory group. As mentioned above in this document, we plan to perform further analysis on the system alone as well as on the bioavailability offered by these applications. We, however have previously employed most of these methods in other reports from our group.
Point 6. It is advised to include the drug release profile of developed formulation system through dialysis bag technique and/or everted-gut sac study to determine permeability profile of developed curcumin nanoemulsion system.
Response 6:
we appreciate your suggestion; however, as much as we would like to add more data for this article, we believe that such information will be better suited for a specific upcoming paper, that will also cover other aspects. As it has been mentioned, this research is based on the hepatoprotective effect of curcumin using a nanoemulsion system by stabilizing it with an original, non-commercial emulsifier, which confers multiple benefits.
Point 7. The author has mentioned in his conclusion that the developed system is capable of increasing the bioavailability of curcumin, therefore, it is suggested to include the pharmacokinetic profile of the developed curcumin nanoemulsion system in a suitable animal model to confirm the robustness/superiority of the developed formulation system compared to the previous report.
Response 7:
We appreciate this observation; however, for this article the focus was not placed on determining curcumin bioavailability. Nevertheless, in our research group assays to elucidate curcumin bioavailability have been previously performed. Here we add the resulting published papers that address this observation, despite them not using this exact system, it is shown that the aim was to increase the bioavailability and bioactivity of curcumin through the use of nanocarrier systems by modifying the emulsifiers and not using commercial ones. It should be noted that by modifying our emulsifiers, multiple benefits are conferred to the system, not only in terms of stability, but also in biological aspects.
- Esperón-Rojas, A.A.; Baeza-Jiménez, R.; Santos-Luna, D.; Velasco-Rodríguez, L. del C.; Ochoa-Rodríguez, L.R.; García, H.S. Bioavailability of curcumin in nanoemulsions stabilized with mono- and diacylglycerols structured with conjugated linoleic acid and n-3 fatty acids. Biocatal. Agric. Biotechnol. 2020, 26, doi:10.1016/j.bcab.2020.101638.
- Ochoa-Flores, A.A.; Hernández-Becerra, J.A.; Cavazos-Garduño, A.; Soto-Rodríguez, I.; Sanchez-Otero, M.G.; Vernon-Carter, E.J.; García, H.S. Enhanced Bioavailability of Curcumin Nanoemulsions Stabilized with Phosphatidylcholine Modified with Medium Chain Fatty Acids. Curr. Drug Deliv. 2017, 14, 377–385, doi:10.2174/1567201813666160919142811.
- Chávez-Zamudio, R.; Ochoa-Flores, A.A.; Soto-Rodríguez, I.; Garcia-Varela, R.; García, H.S. Preparation, characterization and bioavailability by oral administration of O/W curcumin nanoemulsions stabilized with lysophosphatidylcholine. Food Funct. 2017, 8, 3346–3354, doi:10.1039/c7fo00933j.
Also, the word "bioavailability" has been removed from the conclusion section to avoid misunderstandings, leaving only "bioactivity" in line 516.
Reviewer 3 Report
Comments:
This paper deals with the preparation and characterization of curcumin Nanoemulsions stabilized with MAG & DAG-MCFAs for the treatment of hepatic steatosis and the authors evaluate the Nanoemulsions efficacy in murine model. In my opinion this is an interesting research paper because Nanoemulsions are attracting a lot of interest nowadays, but the article cannot be accepted in this form but only after major revision.
I suggest major points:
The introduction lacks one large paragraph regarding the features of nanoemulsions, their advantages respect to other drug delivery systems and the research hypothesis because and how they were used. Moreover, it is well known that there are some difficulties in distinguish nanoemulsions from microemulsions. It’s important that the authors take in account this problem.
Authors should consider including few references in this regard:
- Nano-emulsions and Micro-emulsions: Clarifications of the Critical Differences Nicolas Anton & Thierry F. Vandamme Pharm Res (2011) 28:978–985 DOI 10.1007/s11095-010-0309-1;
- Soft Matter 12 (2016) 28269;
- Nanoemulsions of Satureja montana Essential Oil: Antimicrobial and Antibiofilm Activity against Avian Escherichia coli Strains, Pharmaceutics 2021, 13, 134. https://doi.org/10.3390/pharmaceutics13020134
Design experiments:
Ternary phase diagram construction is the best way to observe the dispersion formation by mixing of Nanoemulsions components. This study is important to select the optimized amounts of components in the development of Nanoemulsions. Is it possible for the NE with MAG and DAG MCFAs to prepare pseudoternary diagram?
More appropriate knowledge of the Nanoemulsions state of the art and characterization studies of drug delivery systems are necessary. I think that different experiments could be useful to better characterize nanocarriers for the in vivo application.
I suggest only a few examples of experiments:
- ζ potential by DLS in order to explain Nanoemulsion stability;
- Nanoemulsions stability evaluation in different biological media and in different external conditions (e.g. different pH according to supposed administration route);
- Nanoemulsions characterization in terms of size and ζ potential at 37°C;
- Nanoemulsion morphological studies.
Is it possible to evaluate differences features between empty and loaded Nanoemulsions?
Moreover, in the experiments results must be included statistical significance wherever possible and, in the paper, more recent references must be added.
Author Response
Response to Reviewer 3 Comments
This paper deals with the preparation and characterization of curcumin Nanoemulsions stabilized with MAG & DAG MCFAs for the treatment of hepatic steatosis and the authors evaluate the Nanoemulsions efficacy in murine model. In my opinion this is an interesting research paper because Nanoemulsions are attracting a lot of interest nowadays, but the article cannot be accepted in this form but only after major revision.
Point 1. The introduction lacks one large paragraph regarding the features of nanoemulsions, their advantages respect to other drug delivery systems and the research hypothesis because and how they were used. Moreover, it is well known that there are some difficulties in distinguish nanoemulsions from microemulsions. It’s important that the authors take in account this problem.
Response 1:
We appreciate your suggestion, we have expanded the introduction section with the required information, it is on line 89 of the manuscript.
“Nanoemulsions (NE) are transparent/translucent heterogeneous systems composed of a continuous phase and a dispersed phase in the form of droplets [22,23], characterized by particle sizes between 20-200 nm; however, not all authors agree on this range, this is because the size is considered to vary depending on the source, without being tightly specified [24]. Despite the droplet size, the differences between emulsion, microemulsion and NE often generate confusion; NE mainly differ from emulsions in terms of gravitational stability, i.e. the smaller the size the greater the stability; evidence also shows that bioavailability of lipophilic compounds increases when globule sizes are <200 nm. Additionally, NE differ from microemulsions, because of their smaller size and thermodynamic stability; nonetheless, they are adversely affected by changes in temperature and composition; therefore, NE are widely used in the food and pharmaceutical industries, even though they are thermodynamically unstable, meaning that eventually the phases will separate because of free energy difference [24-26]. To improve the stability of NE, structured emulsifiers such as monoacylglycerides (MAG) and diacylglycerides (DAG) enriched with medium-chain fatty acids (MCFAs) have been proposed, which not only provide stability to the system, but also confer health benefits to people with several metabolic disorders [27]; these emulsifiers are considered non-ionic small surfactant molecules commonly employed in food emulsions, cosmetic, pharmaceutical emulsifiers. These acylglycerides are commonly produced by chemical processes using materials such as fats, oils and glycerol; that provide options to select the length of the chains that form them, providing a high impact on their functionality [28], an example is CAPMUL® which can be used as a primary solubilizer or as an emulsifier. MAG and DAG are produced by interesterification of oils with glycerol, forming a mixture of MAG, DAG, TAG, glycerol and free fatty acids [28]; thus, allowing for better contact of the substrates without the need of a solvent [29].”
The main objective of this research was to study the hepatoprotective effect of curcumin in a NE system, stabilized and enhanced with a non-commercial emulsifier of mono- and diacylglycerides enriched with medium chain fatty acids, on an induced hepatic steatosis by high fructose intake, on a Wistar rat model.”
Point 2. Authors should consider including few references in this regard:
Nano-emulsions and Micro-emulsions: Clarifications of the Critical Differences Nicolas Anton & Thierry F. Vandamme Pharm Res (2011) 28:978–985 DOI 10.1007/s11095-010- 0309-1; Soft Matter 12 (2016) 28269; Nanoemulsions of Satureja montana Essential Oil:
Antimicrobial and Antibiofilm Activity against Avian Escherichia coli Strains, Pharmaceutics 2021, 13, 134. https://doi.org/10.3390/pharmaceutics13020134
Response 2:
We appreciate the reference you suggested to complement our work, it can now be found in the introduction section on lines 93-101 of the manuscript.
“Nanoemulsions (NE) are transparent/translucent heterogeneous systems composed of a continuous phase and a dispersed phase in the form of droplets [22,23], characterized by particle sizes between 20-200 nm; however, not all authors agree on this range, this is because the size is considered to vary depending on the source, without being tightly specified [24]. Despite the droplet size, the differences between emulsion, microemulsion and NE often generate confusion; NE mainly differ from emulsions in terms of gravitational stability, i.e. the smaller the size the greater the stability; evidence also shows that bioavailability of lipophilic compounds increases when globule sizes are <200 nm. Additionally, NE differ from microemulsions, because of their smaller size and thermodynamic stability; nonetheless, they are adversely affected by changes in temperature and composition; therefore, NE are widely used in the food and pharmaceutical industries, even though they are thermodynamically unstable, meaning that eventually the phases will separate because of free energy difference [24-26]. To improve the stability of NE, structured emulsifiers such as monoacylglycerides (MAG) and diacylglycerides (DAG) enriched with medium-chain fatty acids (MCFAs) have been proposed, which not only provide stability to the system, but also confer health benefits to people with several metabolic disorders [27]; these emulsifiers are considered non-ionic small surfactant molecules commonly employed in food emulsions, cosmetic, pharmaceutical emulsifiers. These acylglycerides are commonly produced by chemical processes using materials such as fats, oils and glycerol; that provide options to select the length of the chains that form them, providing a high impact on their functionality [28], an example is CAPMUL® which can be used as a primary solubilizer or as an emulsifier. MAG and DAG are produced by interesterification of oils with glycerol, forming a mixture of MAG, DAG, TAG, glycerol and free fatty acids [28]; thus, allowing for better contact of the substrates without the need of a solvent [29].”
Point 3. Design experiments:
Ternary phase diagram construction is the best way to observe the dispersion formation by mixing of Nanoemulsions components. This study is important to select the optimized amounts of components in the development of Nanoemulsions. Is it possible for the NE with MAG and DAG MCFAs to prepare pseudoternary diagram?
Response 3:
As mentioned in this document, the main objective of this research was to prove the hepatoprotective activity of curcumin by placing it in a nanoemulsified system with the benefit provided by mono- and diacylglycerides enriched with medium chain fatty acids, part of our group has been responsible for conducting the relevant tests to determine that it is a system with good stability and a monomodal dispersion; however, we included some articles published by our group where you can see this type of diagram.
Bravo-Alfaro, D.A.; Muñoz-Correa, M.O.F.; Santos-Luna, D.; Toro-Vazquez, J.F.; Cano-Sarmiento, C.; García-Varela, R.; García, H.S. Encapsulation of an insulin-modified phosphatidylcholine complex in a self-nanoemulsifying drug delivery system (SNEDDS) for oral insulin delivery. J. Drug Deliv. Sci. Technol. 2020, 57, 101622, doi:10.1016/j.jddst.2020.101622.
Muñoz-Correa, M.O.F.; Bravo-Alfaro, D.A.; García, H.S.; García-Varela, R. Revista Mexicana de I ngeniería Química. Rev. Mex. Ing. Quím. 2013, 12, 505–511.
Point 4. More appropriate knowledge of the Nanoemulsions state of the art and characterization studies of drug delivery systems are necessary. I think that different experiments could be useful to better characterize nanocarriers for the in vivo application.
I suggest only a few examples of experiments:
- ζ potential by DLS in order to explain Nanoemulsion stability.
- Nanoemulsions stability evaluation in different biological media and in different external conditions (e.g. different pH according to supposed administration route);
- Nanoemulsions characterization in terms of size and ζ potential at 37°C;
- Nanoemulsion morphological studies.
Is it possible to evaluate differences features between empty and loaded Nanoemulsions?
Response 4:
The objective of this paper was not to determine the stability and/or characterize the nanoemulsion because those analyses have already been previously performed (see DOI: 10.1080/10242422.2019.1646254 DOI: 10.5650/jos.ess17010) which, as already mentioned, are contributions from our group. We reproduced those results starting from a predetermined system with the best conditions established to confirm their effects.
Point 5. Moreover, in the experiments results must be included statistical significance wherever possible and, in the paper, more recent references must be added.
Response 5:
Modifications were made to section 3.2. NE of curcumin with MAG and DAG in line 251 of the manuscript.
“3.2. Curcumin NE with MAG and DAG
Emulsions were prepared in a 97:3 water-oil phase ratio with 1% emulsifier and 1% Kolliphor® EL to dissolve curcumin as reported by Esperón-Rojas et al. (2020) [27]; a modification to the proposed methodology was the sonication time; it was observed in preliminary runs, that subjecting the sample to 3 cycles instead of 2 was favorable in decreasing particle size, this variation produced a particle size of 184.4 nm ± 1.02 nm. Comparison of particle size and PDI of this work with other investigations using MCFAs are presented in Table 2.”
Due to the suggestions, we had from all the reviewers, we added updated references concerning nanoemulsions and the effect of curcumin in carrier systems; bibliography insertion is at the end of the document in the "References" section starting on line 533.
Round 2
Reviewer 1 Report
The revised version of the manuscript looks much better. I accept that paper for publication.
Reviewer 2 Report
The manuscript (pharmaceutics-1151465) entitled "Effect of curcumin nano-emulsions stabilized with MAG & DAG-MCFAs in a fructose-induced hepatic steatosis rat model" improved well and given suggestion nicely incorporated by the authors. Therefore, the present version of the manuscript should be considered for publication in PHARMACEUTICS.
Reviewer 3 Report
Thanks to the authors for editing the text following the suggestions.